# Translational Insights and New Therapeutic Perspectives in Head and Neck Tumors

**DOI:** 10.3390/biomedicines9081045

**Published:** 2021-08-19

**Authors:** Morena Fasano, Francesco Perri, Carminia Maria Della Corte, Raimondo Di Liello, Giuseppina Della Vittoria Scarpati, Marco Cascella, Alessandro Ottaiano, Fortunato Ciardiello, Raffaele Solla

**Affiliations:** 1Medical Oncology, Department of Precision Medicine, University of Campania “Luigi Vanvitelli”, 80138 Naples, Italy; morenafasano@ymail.com (M.F.); carminiamaria.dellacorte@unicampania.it (C.M.D.C.); raimondo.diliello@unicampania.it (R.D.L.); fortunato.ciardiello@unicampania.it (F.C.); 2Medical and Experimental Head and Neck Oncology Unit, Istituto Nazionale Tumori IRCCS Fondazione Pascale-IRCCS di Napoli, Via M. Semmola, 80131 Naples, Italy; 3Medical Oncology Unit, Hospital Sir Apicella, Pollena Trocchia, 80040 Naples, Italy; giuseppina.dellavittoria@gmail.com; 4Division of Anesthesia, Istituto Nazionale Tumori IRCCS Fondazione G. Pascale, 80100 Naples, Italy; m.cascella@istitutotumori.na.it; 5SSD Innovative Therapies for Abdominal Metastases, Istituto Nazionale Tumori IRCCS Fondazione G. Pascale, 80100 Naples, Italy; a.ottaiano@istitutotumori.na.it; 6Italian National Research Council, Institute of Biostructure & Bioimaging, 80131 Naples, Italy; raffaelesolla@yahoo.com

**Keywords:** head and neck squamous cell carcinoma, immunotherapy, DNA damage response, epithelial growth factor receptor

## Abstract

Head and neck squamous cell carcinoma (HNSCC) is characterized by a high mortality rate owing to very few available oncological treatments. For many years, a combination of platinum-based chemotherapy and anti-EGFR antibody cetuximab has represented the only available option for first-line therapy. Recently, immunotherapy has been presented an alternative for positive PD-L1 HNSCC. However, the oncologists’ community foresees that a new therapeutic era is approaching. In fact, no-chemo options and some molecular targets are on the horizon. This narrative review addresses past, present, and future therapeutic options for HNSCC from a translational point of view.

## 1. Introduction

As most cases of head and neck squamous cell carcinoma (HNSCC) are diagnosed in locally advanced or metastatic settings, this set of oncological disease is characterized by a high mortality rate. Unfortunately, in the recurrent/metastatic setting, the only available treatments are represented by systemic treatments and palliative radiotherapy and/or surgery [1]. Various topical drugs have been proposed as palliative treatments in the management of HNSCC as an adjuvant or neoadjuvant therapy, with controversial results [2,3].

Many efforts have been pursued in identifying new targets and innovative therapies to increase therapeutic options over chemotherapy, as well as to improve response rates, survival, and quality of life. This paper is aimed at dissecting the therapeutic pathway, which led to the discovery of targeted therapy. In particular, the role of immunotherapy that currently represents “the present” is addressed [4]. Finally, the crucial importance of translational research to prepare new therapies tailored on patient and tumor characteristics is underlined (see Appendix A for the methodology applied to search literature).

## 2. The Past and Present: Targeting EGFR

### 2.1. EGFR mAb

EGFR (epithelial growth factor receptor) is a member of the ErbB/HER family, overexpressed in about 90% of HNSCCs. Its expression is related to poor survival because of resistance to radiotherapy and local treatments; nevertheless, its prognostic role is still quite controversial [5]. In 2006, cetuximab, an anti-EGFR monoclonal antibody (mAb), was approved in HNSCC treatment. Cetuximab is an IgG1 monoclonal antibody that blocks EGFR activation by specifically binding to the extracellular domain of EGFR, thus inducing EGFR internalization and downregulation. Inhibition of EGFR-downstream pathways is able to interfere with cancer growth. Moreover, cetuximab has anti-body-dependent cell-mediated cytotoxicity, owing to its IgG1 isotype, as it directs cytotoxic immune-cells against EGFR-expressing tumor cells [6,7]. In recurrent/metastatic HNSCC (R/M HNSCC), cetuximab was approved in combination with platinum-based chemotherapy, showing fairly good results in terms of overall survival, response rate, and progression-free survival [8].

Previously, other EGFR mAbs were tested without reaching clinical approval. In the SPECTRUM trial [9], panitumumab, a fully human mAb, in combination with polychemotherapy (based on platinum + fluorouracil), did not show any benefit in the metastatic setting. The difference in clinical activity between cetuximab and panitumumab was probably due to their different isotype conformation with a consequent heterogeneous induction of antibody-dependent cellular cytotoxicity (ADCC) (IgG1 for cetuximab and IgG2 for panitumumab). Similarly, zalutumumab is an IgG1 mAb that can block EGFR and induce ADCC, but it did not increase the clinical outcome in the same setting. In an exploratory, open-label, randomized, multicenter study in operable HNSCC patients, imgatuzumab (GA201), which is another primatized glycol-engineered IgG1 mAb with ADCC-related immune effects, showed promising results in terms of tumor immune infiltration. Sym004, a new generation anti-EGFR mAb, displayed only modest anti-tumor activity in a proof of concept trial, without further clinical exploration [10].

Losatuxizumab vedotin (ABBV-221) is a second-generation antibody–drug conjugate (ADC) anti-EGFR that obtained some responses in a multicenter phase 1 study also enrolling patients with HNSCC, but it was poorly tolerated for the high frequency of infusion reactions [11].

### 2.2. EGFR TKIs

In addition to EGFRmAb, small-molecule tyrosine kinase inhibitors (TKIs) that bind to the intracellular domain of EGFR were studied.

Two first-generation reversible EGFR-TKIs, erlotinib and gefitinib, were tested in multiple clinical trials without obtaining any benefit over the EXTREME regimen [12,13]. However, a phase II study evaluating the combination of erlotinib, carboplatin, paclitaxel, and cetuximab in patients with metastatic or recurrent HNSCC is ongoing. Nevertheless, a preliminary analysis of the first 24 patients has shown that overall response rate (ORR), progression-free survival (PFS), and overall survival (OS) were similar to historical data obtained with the EXTREME regimen [14]. In trials enrolling patients with both metastatic and locoregional disease, dacomitinib, a second-generation irreversible EGFR TKI, led to inconclusive results [15].

Different data were obtained with the other irreversible second-generation TKI inhibitor, afatinib, which significantly improved PFS, in a phase III LUX-head and neck 3 trial, as second-line treatment when compared with the standard methotrexate [16].

EGFR mutations often confer greater responsiveness to small anti-EGFR molecules. Nevertheless, the EGFR gene is very rarely mutated in HNSCC, as is the case for lung cancers. This paramount gap can explain the unconvincing results obtained with EGFR inhibitors in various clinical studies. Despite the conflicting results obtained with the use of different anti EGFR therapies, EGFR still remains a key therapeutic target. Therefore, even if most of these drugs are not approved in clinical practice, we believe that these positive data confirm the biological importance of EGFR signaling in HNC development. The study of the EGFR pathway in HNSCC deserves further scientific efforts.

## 3. The Present and Future: Immunotherapy

### 3.1. Check-Point Inhibitors (ICIs)

Recently, immunotherapy has greatly modified the therapeutic standard of HNSCC in the R/M setting. The immune system plays a fundamental role in regulating tumor growth, as several solid and hematological tumors develop more easily in immunocompromised individuals, thus underlining the importance of “immunological surveillance” against the growth of tumor cells. On this rationale, many efforts have been focused on the development of immunotherapy drugs to restore the ability of the immune system for detecting and destroying cancer cells [17].

Specifically, progress in the understanding of mechanisms regulating the immune system activity has shed light on the crucial role of many proteins and lymphocytes. In particular, antigens associated with cytotoxic T lymphocytes-4 (CTLA-4), programmed cell death ligand 1 (PDL-1), and indoleamine-2,3-dioxygenase (IDO), as well as lymphocytes with regulatory functions (T-regulatory cells–Tregs) and myeloid-derived suppressive cells (MDSC), can profoundly modulate the immune response and belong to the so-called “immune checkpoints”, offering new therapeutic strategies.

Recently, published data have clearly indicated that immunotherapy (anti-CTLA-4 and anti-PD-1/PD-L1 antibodies) represents an important therapeutic option for HNSCC patients. However, despite promising treatment results, a significant number of patients still fail to achieve clinically meaningful benefits. For this reason, in the era of precision medicine, identifying reliable predictive factors to select patients who are most likely to benefit from treatment with immune agents is a crucial and open challenge in oncology [18].

### 3.2. CTLA-4

CTLA-4 is the first checkpoint receptor that has been successfully studied and tested in cancer as a target [19]. It has a critical role in maintaining activation of T cells, as demonstrated by the lethal systemic immune-hyperactivation phenotype of CTLA-4 knockout mice [20]. Currently, two human anti-CTLA-4 antibodies are used and studied in clinical practice in solid tumors. Ipilimumab, approved in the treatment of advanced melanoma, and tremelimumab, under development in several solid tumors, works by binding to CTLA-4 and blocking its immunosuppressive signal. As a result, activated T cells, including those activated by tumor antigens, can continue to proliferate, producing cytokines and exerting their cytotoxic effector functions in the tumor microenvironment. The first data came from a case report. In a 46-year-old male with relapsed PDL1 positive HNSCC, Schwab et al. [21] showed that the combination ipilimumab plus nivolumab induced a partial response after 8 weeks from the start of treatment and complete response after 4 months of therapy.

### 3.3. Programmed Death-1 (PD-1/PDL-1)

Another immune checkpoint receptor studied is the programmed death receptor 1 (PD1). It is a member of the CD28 superfamily that delivers negative signals upon interaction with its two ligands, the programmed cell death ligand 1 (PD-L1) and the programmed cell death ligand 2 (PD-L2). Similar to CTLA-4, PD-1 plays a key role in regulating and maintaining the balance between T cell activation and in promoting self-tolerance. Unlike CTLA-4, PD-1 is widely expressed and can be found not only on the surface of T cells, but also on that of B and NK (natural killer) cells. While CTLA-4 mainly regulates the activation of T cells in lymphatic tissues, the primary role of PD-1 is to suppress the inflammatory activity of T cells in peripheral tissues during a cell-mediated or inflammatory immune response. In turn, targeting PD-1/PD-L1 can produce a wide-ranging effect. PD-L1 ligand is commonly upregulated on several human solid tumors, including HNSCC. Consequently, it represents a biomarker in clinical practice (such as, for example, in lung cancer) [22].

The expression of PD-L1 on tumor cells, as assessed by immunohistochemistry (IHC), was initially identified as a biomarker for predicting response to treatment with anti-PD-1/anti-PD-L1 therapies. This topic has been widely studied on different types of cancer with mixed results [23,24].

The factors predictive for a good response to anti-PD1 treatment are not fully understood. Indeed, PDL1 expression is only one of the potential determinants of immunotherapy efficacy. The lack of benefit in some patients with PD-L1 positive cancer implies that other molecular mechanisms are involved in resistance to checkpoint inhibition. It was also demonstrated that, in HNSCC, the combined positive score (CPS), calculated as the number of PD-L1 positive cells including tumor, lymphocytes, and macrophages, in relation to total tumor cells, appears to be more specific than the tumor proportional score (TPS). The latter measures PD-L1 expression only on tumor cells, in the selection of patients with HNSCC who may benefit from immunotherapy treatment. This feature was confirmed by the results of the first-line KEYNOTE-048 study. It demonstrated the superiority of the anti PDL-1pembrolizumab, alone or in combination with chemotherapy (cisplatin or carboplatin and 5FU), over the standard platinum-5-fluorouracil-cetuximab regimen, in patients with HNSCC PDL-1 CPS positive (CPS > 1) [25]. Currently, pembrolizumab, an anti-PD-L1 mAb, alone or in combination with platinum-based chemotherapy, represents the new standard of care in first-line therapy for HNSCC with CPS ≥ 1.

The positive response to immune checkpoints inhibitor can be also predicted by the presence of other biomarkers such as the expression of PD-L2, the other PD-1 ligand, as emerged from the results of the KEYNOTE-012 study [26]. In HNSCC, PD-L1 is overexpressed in about 50–60% of cases; therefore, PD-1/PD-L1 and PD-L2 inhibitors may represent the main class of immunotherapy drugs for this cancer type.

## 4. Immunotherapy Biomarkers

### 4.1. TMB

A large portion of HNSCC has a high tumor mutational burden (TMB). It is related to heavy cigarette smoking (typical of patients with HNSCC) and to the presence of human papillomavirus (HPV) and Epstein–Barr virus (EBV) viral infections. It is likely that tumors with a large number of somatic genome mutations develop a higher specific T cell response to tumor neoantigens, which results in greater susceptibility to immunotherapy. For this reason, TMB has been proposed as a new biomarker of response to immune agents. Several studies have explored the correlation between elevated TMB and the benefit of anti-CTLA-4 and anti-PD-1 antibodies, and mutational load has proved to be a very promising biomarker as tumors with TMB > 100 somatic mutations associated with an increase in survival [27]. Clinical findings in the HNSCC cohort of the KEYNOTE-012 study showed that patients with both elevated TMB and high PD-L1 expression responded to treatment with pembrolizumab; moreover, there was no direct association between TMB and PD-L1 expression. This confirms that TMB and PD-L1 are two independent biomarkers for the prediction of the response to immunotherapy [28]. Recently, the study published by Zhang et al. [29] found that high levels of TMB are also associated with poor prognosis, advanced stage, and large primary tumor size in HNSCC patients.

### 4.2. Microsatellite Instability

Microsatellite instability (MSI) refers to a specific “hypermutator phenotype” corresponding to the presence of somatic or inherited DNA mismatch repair genes mutations. In routine use, detection of MSI is done by IHC for MMR (mismatch repair) proteins or by DNA profiling. MSI-high is associated with the efficacy of PD-1 blockade in other tumor types [30]; however, the incidence of the MSI-high phenotype is very low in HNSCC and no data are available for using it in clinical practice [31].

### 4.3. New Immunotherapy Biomarkers and Targets

Other emerging biomarkers studied to assess the primary response to immunotherapy are tumor-infiltrating lymphocytes (TILs), HPV, IDO, inducible T-cell co-stimulator (ICOS), and NKG2A (natural killer group 2A) receptor. The role of TIL as a predictor for patient selection has been studied in various tumor types. It has been observed that, in tumor samples obtained after immunotherapy, a high density of TILs is associated with increased activity of these drugs [32]. Furthermore, it was also found that a better response rate to pembrolizumab has been observed in melanoma patients with higher CD8+ density during treatment [33]. In the cohort study published by Spector et al. [34], TILs’ levels were an independent prognostic factor in patients with HNSCC.

Regarding viral infections, HPV positivity correlates with a better clinical outcome with immunotherapy, thus representing a favorable clinical prognostic biomarker in HPV-positive disease [34]. Chen et al. [35] demonstrated that p16 protein expression is highly correlated with PD-L1 expression in HNSCC samples, thus explaining why these tumors probably respond better to anti-PD-1/PD-L1 drugs. Some evidence suggests that HPV positivity is predictive of response to anti-PD1 agents. In this regard, the KEYNOTE-012, CheckMate 141, and the KEYNOTE 048 investigations showed an improvement in outcome in HPV-positive patients compared with HPV-negative patients. Despite this, HPV positivity cannot currently be considered to select patients for immunotherapy [36]. IDO plays an important role in immunity as it intervenes in the natural defense against various pathogens; it is produced in response to inflammatory stimuli and performs an immunosuppressive function by limiting the activity of T lymphocytes, on the one hand, and activating the mechanisms involved in immune tolerance, on the other hand [37]. Retrospective studies in patients with HNSCC showed that high levels of IDO expression are correlated with worse outcomes and a poorer prognosis, probably owing to the direct association of IDO with regulatory T cells (T-Reg). Although evidence from other tumors, such as melanoma, did not show improvement in outcome in the group of patients treated with IDO inhibitor epacadostat, some studies have been conducted in patients with HNSCC. In the phase I/II study ECHO-202/Keynote 037, which evaluated the combination of epacadostat plus pembrolizumab, in the two patients enrolled with HNSCC, disease stability was obtained as the best response with a disease response of 34% and a disease control rate of 39% [38]. Instead, the results of the phase III Keynote 669/Echo 304 study that evaluated the combination of epacadostat plus pembrolizumab versus pembrolizumab monotherapy versus the EXTREME regimen are awaited. A phase II study evaluating the combination of BMS-986205, an IDO1 inhibitor with nivolumab (NCT03854032) in stage II–IV patients with HNSCC, is still ongoing. Another molecule involved in immunity and being studied is ICOS, a protein stimulated by both the T cell receptor and CD28 signals. A potential therapeutic strategy to overcome resistance to anti PD1/PDL-1 could be represented by the combination of anti PD1 antibodies with the ICOS agonists. In this regard, a double-blind, randomized phase 3 study is underway evaluating the combination of an ICOS agonist, GSK3359609, with pembrolizumab versus placebo plus pembrolizumab in first-line treatment in patients with HNSCC R/M PD-L1 positive [39]. Lastly, an alternative target of the immune checkpoint could be represented by the NKG2A receptor, present on the surface of NK cells and on CD8 + T lymphocytes. Monalizumab is an IgG4 class antibody whose function is precisely to block NKG2A by promoting antitumor immunity and increasing antibody-dependent cell-mediated cytotoxicity. As the combination of monalizumab plus cetuximab showed promise in a phase II study (with a 31% response rate), a phase III study is ongoing to evaluate the combination of the two antibodies in patients with platinum-resistant R/M SHCCN previously treated with immunotherapy [40].

## 5. What Is New for the Future: IO Combinations

### 5.1. Clinically Relevant Molecular Alterations in HNSCC

The development of new technologies helped to dissect tumor genomic molecular alterations, thus identifying novel therapeutic targets. How to translate these molecular features into clinically relevant treatments options is still not clear. The level of evidence of actionable mutations in HNSCC on the basis of the European Society for Medical Oncology (ESMO) Scale for Clinical Actionability of Molecular Targets (ESCAT) [41] is an instrument to help oncologists in selecting treatments. Therefore, alterations of 33 genes have been studied. Among these alterations, HRAS-activating mutations (targetable by tipifarnib, a farnesyltransferase inhibitor) and similarly for NTRK (neurotrophic tyrosine kinase receptor) fusions, seem to be very interesting. These alterations have also been proposed as new targets in combination with immunotherapy [42]. Based on positive results of palbociclib (CDK4/6 inhibitor) and afatinib in molecular subgroups from a retrospective investigation [43], CDKN2A-inactivating alterations and EGFR amplification have been ranked in a high position.

EMERGING TARGETS: MEK. ErbB family proteins like EGFR, HER2, HER3, and HER4 play important roles in many cancer types, including head and neck. Despite the few goals achieved in targeted drug development, many translational and preclinical experiences have studied the relation between ErbB proteins and drug sensitivity in HNSCC. Afatinib, an irreversible inhibitor of EGFR, HER2, and HER4, was studied in combination with the MEK inhibitor PD0325901, with the aim to inhibit cisplatin-resistant HNSCC cells lines. Afatinib was shown to inhibit the Akt/mTOR activity and to promote the phosphorylation of EGFR, HER2, and HER3, concomitantly with an up-regulation of MEK/ERK signaling. More interestingly, MEK inhibitor PD0325901 blocked ERK phosphorylation, while the combination inhibited if all these pathways synergistically [44].

Recently, MEK inhibition has also been demonstrated to overcome the limited efficacy of CDK4/6 inhibitor. In fact, Fang et al. [45] reported that treatment with trametinib (MEK inhibitor) plus palbociclib (CDK4/6 inhibitor) resulted in a G0/G1 cell cycle arrest and apoptotic cell death in HNSCC cells, along with a remarkable decrease of MAPK pathway activation. These results have been confirmed in studies conducted on xenograft mouse models [46].

*DDR.* DNA damage response (DDR) is a cellular process used to report the presence of DNA damage [46]. Therefore, targeting DDR is emerging as a promising therapeutic option in many cancer types, especially where platinum and/or radiotherapy (both acting on DNA damage) are milestones of treatment. In this scenario, many DDR inhibitors are also under investigation in HNSCC that could be considered a prototype of DDR-sensitivity [47].

### 5.2. PARP (Poly ADP-Ribose Polymerases)

In vitro studies demonstrated a high sensitivity of HNSCC (both homologous recombination (HR)-deficient and -proficient) to the radiosensitizing activity of PARP inhibitors (PARPi) [48]. Moreover, other preclinical and clinical experiences demonstrated that PARPi sensitize cancer cells (including HNSCC) to platinum-based chemotherapy, temozolamide, and topoisomerase inhibitors [49,50]. These promising synergistic effects are currently tested in different ongoing clinical trials that combine CT/RT with PARPi (e.g., NCT01758731, NCT01460888, NCT02308072). In addition, other combination strategies of PARPi plus other-than-PARPDDR inhibitors (e.g., CHK1 and WEE1 inhibitors) are also under evaluation in HNSCC [51].

### 5.3. DNA-PK (DNA-Dependent Protein Kinase, Catalytic Subunit)

Different DNA-PK inhibiting molecules have been developed so far. Unfortunately, most showed several pharmacokinetics issues or an unacceptable safety profile [51,52]. As with other DDR inhibitors, DNA-PK development is mainly based on combination strategies, considering that monotherapy showed only modest effects [53]. In general, cells with a defective DNA-PK activity (also artificial) are highly sensitive to radiotherapy, indicating a potential radiosensitizing activity, later confirmed in different preclinical studies [54,55]. Specifically, the radiosensitizing effect of the DNA-PK inhibitor NU7411 was confirmed in preclinical studies in different cancer types such as lung, liver, and breast cancer [56,57]. On these bases, also the combination of EGFR inhibition (involved in the DNA-PK pathway) has been studied, showing an increased radiosensitizing effect in EGFR overexpressing cells and leading to an interesting new research field of the EGFR/DNA-PK co-inhibition [58]. All these promising effects of DNA-PK inhibitors are also under investigation in the clinical setting, as multiple clinical trials in solid tumors are ongoing (not specific for HNC).

### 5.4. ATM/ATR

ATM (ataxia-telangiectasia mutated) and ATR (ataxia-telangiectasia and Rad3 related) play a critical role in cell cycle regulation and DDR, specifically through CHK1 and CHK2 phosphorylation [59]. In HNC, 4–10% and 1–16% of the cases are characterized by ATR and ATM somatic mutations, respectively [60]. As with other DDR inhibitors, ATR/ATM targeting agents showed chemotherapy- and radiotherapy-sensitizing effects that led to preliminary clinical experience as monotherapy or in combinations [61]. M6620 (previously VX-970) is a first-in-class ATR inhibitor currently under investigation in a phase 1 trial in HPV-negative HNSCC (NCT02567422). AZD6738 is another selective ATR inhibitor that was recently demonstrated to enhance radiotherapy response in both HPV-negative and HPV-positive HNSCC in vitro [62]. A clinical trial of AZD6738 plus olaparib is currently ongoing in HNC (NCT02264678), and another biomarker-based study has recently been completed (NCT03022409).

### 5.5. CHK1/2

CHK1, alone or through the recruitment of RAD51, along with CHK2 (and its interaction with p53), are the main components of the DDR system [63,64]. Considering that many preclinical studies confirmed the sensitizing effect of CHK1/2 in p53-deficient cells, and that there is a high rate of Tp53 mutation in HNSCC, the CHK1/2 pathway is emerging as a promising potential new DDR inhibitor in this setting [59,65]. Prexasertib, a CHK1/2 inhibitor, was demonstrated to reduce in vitro survival fraction of HNSCC cell lines combined with cisplatin, with or without RT, mainly through the downregulation of NOTCH signaling target genes (*NOTCH1, NOTCH2*, and *NOTCH3*) and their associated ligands (*JAG1, JAG2, SKP2, MAML2*, and *DLL1*). Moreover, a significant tumor growth delay was observed in vivo in both HPV-positive and HPV-negative mouse xenografts treated with prexasertib, cisplatin, and radiotherapy without additional toxicities [66]. A phase 1 clinical trial of prexasertib combined with cisplatin and cetuximab in advanced HNSCC has completed accrual and the results are awaited (NCT02555644).

### 5.6. WEE1

WEE1 inhibition results in the premature entry of cells in the mitosis phase and, as CHK1 inhibitors, this effect is prevalent in p53-deficient cells [67]. Adavosertib (AZD1775) is a first-in-class WEE1 inhibitor, currently under investigation in a late-phase trial in different cancer types. Its activity in HNC was explored in combination strategies with the aim to potentiate multiple chemo- and radiotherapies [68]. The triplet combination of adavosertib, cisplatin, and docetaxel has been shown to be safe and tolerable in a phase 1 clinical trial in neoadjuvant HNC patients [69]. In addition, as stated for other DDR inhibitors, several shreds of evidence suggest the hypothesis of enhanced activity of these drugs when combined with each other [60]. Indeed, different studies proved, for instance, the synergistic effects of CHK1 and WEE1 inhibitors (e.g., adavosertib plus the CHK1 inhibitor LY2603618) [65] or triplet DDR combinations of PARPi, WEE1, and CHK1 inhibitors [51].

### 5.7. PI3K

Alterations of the PI3K/AKT/mTOR pathway are common in HNSCC with a prevalence of activating mutations of PI3K of 56% and 39% in HPV-positive and HPV-negative HNSCC, respectively [70,71]. Different data support the role of this pathway as an important mechanism of resistance to EGFR inhibitors and RT [72]. Despite these mechanisms, the preclinical model showed that PI3K inhibition alone led to compensatory positive feedback on the RAS/MEK/ERK or EGFR pathway inducing early resistance. On the other hand, combination therapies (e.g., targeting multiple isoforms of PI3K or combining other DDR inhibitors or DNA damaging agents) could achieve synergistic effects [73]. Moreover, as with other targeted therapies in HNSCC, effective biomarkers are still pending. Recently, *NOTCH-1* loss-of-function mutations (*NOTCH1mut*) has shown a potential role as a predictive factor of PI3K/AKT/mTOR inhibition. Thus, in both HNSCC cell lines and xenografts models, *NOTCH1mut* was strongly associated with sensitivity to multiple PI3K/mTOR inhibitors and NOTCH1 inhibition or knockout in wild-type cells increased that effect. However, to overcome all these limitations, pan-PI3K inhibitors (acting on more than one isoform of PI3K) have recently emerged as potential new effective compounds [74]. Currently, buparlisib is the pan-PI3K inhibitor with the most clinical evidence. Buparlisib (BKM120) is an oral reversible PI3K inhibitor that showed anti-proliferative and pro-apoptotic effects in tumor cells, irrespective of the PIK3CA status [75]. Nevertheless, considering early safety data, its use as monotherapy has been replaced with combination strategies [76]. A phase 2 study investigating the combination of buparlisib and cetuximab has been recently completed and the results are awaited (NCT01816984). In addition, the results of a phase 2 study combining buparlisib and paclitaxel showed improved clinical efficacy with a manageable safety profile, suggesting an effective opportunity in pretreated metastatic HNSCC [77], and the phase 3 BURAN trial with this combination is still ongoing (NCT04338399).

### 5.8. CDK

Cyclin-dependent kinases (CDK) play a major role in cell cycle control. In the last years, different CDK4/6 inhibitors have been approved for the treatment of breast cancer and have been tested in late-phase trials in other malignancies [78,79,80]. Recently, CDK inhibition has emerged as a potential mechanism of chemo- and radiosensitization and immune stimulation, leading to preclinical and clinical research that incorporates ICIs and CDK inhibitors in different settings [81].

In HNSCC, beyond CDK4/6 inhibition, other kinases of the same family have been identified as potential biomarkers of response and poor outcome [81,82].

These pieces of evidence also led to the investigation of CDK inhibition in HNSCC in clinical settings. In a phase 1 study in R/M HNSCC, palbociclib plus cetuximab demonstrated a high disease control rate and, in a subsequent phase 2 trial in platinum- or cetuximab-resistant HPV-negative HNSCC, the combination showed efficacy comparable to PD-1 inhibitors and performed better than single-agent cetuximab [83,84]. Despite these early data, recent results from a multicenter phase 2 trial of palbociclib plus carboplatin in the R/M setting did not show improvement in survival outcome and showed that it was associated with significant myelosuppression [85]. Additional clinical trials of CDK inhibitors in HNSCC are ongoing and the results are awaited (NCT03024489, NCT04000529).

Other less frequent molecular alterations. The majority of HNSCCs show a genomic profile consistent with tobacco exposure or, alternatively, are characterized by detectable HPV DNA. Recently, different data have been published on the mutational landscape of HNSCCs showing frequent alterations in TP53, CDKN2A, PTEN, PIK3CA, and HRAS along with mutations in genes related to squamous differentiation as NOTCH1, IRF6, and TP63 [81].

Cancer Genome Atlas profiling on 279 HNSCC cases provided comprehensive genomic sequencing. In HPV-related tumors, *PI3KCA, TRAF3,* and *E2F1* amplifications have been reported as the most frequent alterations, while smoking-related HNSCCs were characterized mostly by *TP53*, *CCND1*, and *CDKN2A* mutations [86]. In the same analysis, beyond these two subgroups that represent the majority of HNSCCs, other types of genomic profiles have been described, related to less prevalent SC that contained inactivating alterations of *NSD1, AJUBA*, and *FAT1* genes (involved in WNT signaling). Distinct profiles were described for tumors arising from the oral cavity. Indeed, *FAT1, CASP8*, *CDKN2A,* and *NOTCH1* mutations were found more frequently in these tumors compared with other HNCs malignancies and other squamous non-HNCs cancers. Another subgroup of tumors of the oral cavity, characterized by a more favorable prognosis, showed infrequent copy number alterations along with activating mutations of or *PIK3* and *HRAS* and, less frequently, mutations of *CASP8*, *NOTCH1*, and *TP53* [86,87].

## 6. Conclusions 

In conclusion, although new biomarkers-driven approaches and new clinical investigations are needed, possible changes in therapeutic scenario of HNSCC can be expected. A modern approach to cancer treatment should include molecular profiling of tumors that can lead to a more personalized approach (in Figure 1, you can see the possible targets that can be “hit” by the various drugs we have available). The therapeutic strategies employed, whether chemotherapy, targeted therapy, or immunotherapy, although effective, are, however, burdened by a still too high percentage of failures, and this is often not easily explained. The study of biomarkers predicting response to immunotherapy, as well as the study of the mutational status of HNSCC, or even the study of some predictive gene polymorphisms of poor or good response to some chemotherapeutic drugs (cisplatin, fluorouracil), can completely subvert the therapeutic scenario. In fact, the early identification of poor-responders as well as good-responders to the various treatments should be the achievable goal in the near future. New clinical investigations are needed to better predict the clinical relevance of tumor molecular alterations and the benefit of targeted therapy/immunotherapy. Table 1 shows the main drugs employed in HNSCC.

## Figures and Tables

**Figure 1 biomedicines-09-01045-f001:**
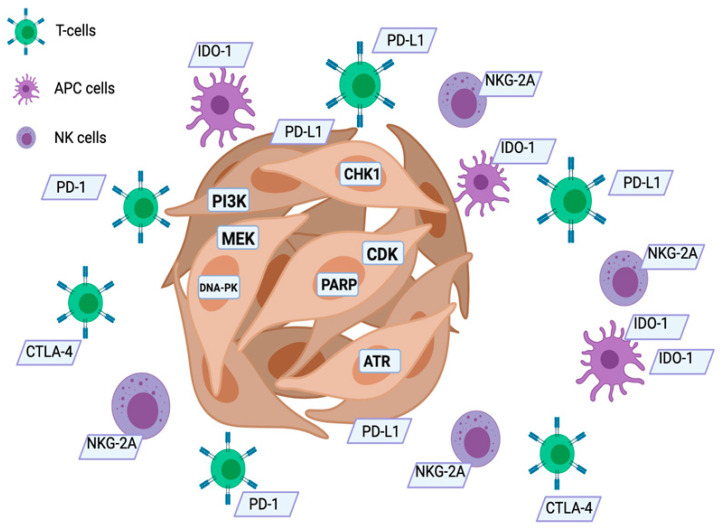
New possible targets in HNSCC. APC: antigen presenting cells; NK: natural killer cells; PARP: poly (ADP-ribose) polymerase; IDO-1: indoleamine 2,3-dioxygenase; ATR: ataxia telangiectasia and Rad3-related protein.

**Table 1 biomedicines-09-01045-t001:** Drugs employed in HNSCC.

Drug	Category	Status
Cetuximab	Targeted Therapy (anti-EGFR)	APPROVED
Panitumumab	Targeted Therapy (anti-EGFR)	experimental
Losatuxizumab vedotin	Targeted Therapy (anti-EGFR)	experimental
Erlotinib	Targeted Therapy (anti-EGFR)	experimental
Gefitinib	Targeted Therapy (anti-EGFR)	experimental
Afatinib	Targeted Therapy (anti-EGFR)	experimental
Nivolumab	Immunotherapy (anti PD-1)	APPROVED
Ipilimumab	Immunotherapy (anti CTLA-4)	experimental
Pembrolizumab	Immunotherapy (anti PD-1)	APPROVED
Atezolizumab	Immunotherapy (anti PD-L1)	experimental
Monalizumab	Immunotherapy (anti-NKG2)	experimental
Palbociclib	Targeted Therapy (anti-CDK 4/6)	experimental
Trametinib	Targeted Therapy (anti-Mek)	experimental
PARP-inhibitors	Targeted Therapy (anti-PARP)	experimental
AZD6738	Targeted Therapy (anti-ATR)	experimental
Prexasertib	Targeted Therapy (anti-*CHK1/2*)	experimental
Adavosertib	Targeted Therapy (anti-*WEE1*)	experimental
Buparlisib	Targeted Therapy (anti-*PI3K*)	experimental
Epacadostat	Immunotherapy (anti-IDO1)	experimental
GSK3359609	Immunotherapy (ICOS agonist)	experimental

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
