# Peer review of "Translational Insights and New Therapeutic Perspectives in Head and Neck Tumors"

_biomedicines, 2021, doi:10.3390/biomedicines9081045_

Round 1

Reviewer 1 Report

A narrative review about current perspectives in head and neck tumor management. I found the article quite interesting to read,; I have some queries:

Although the article is not a systematic review, It would be useful to add a materials and methods section, where authors should highlight what databases were searched (PubMed, Scopus, Google Schiolar) and what keywords were used, even if the subsequent selection of the articles was not sistematical.

I think it should be at least reported in the manuscript's introduction that: " Various topical drugs have been proposed in the management of HNSCC and as an adjuvant or neoaudivant therapy, with controversial results" and cite an article such as : doi: 10.3390/medicina57060563. and doi: 10.3390/curroncol28040213.

What is said in the conclusion paragraph is very interesting...i think the authors should expand it and add their personal considerations. You should also move figure 1 earlier in the text.

Thank You

Author Response

Dear Editor,

We have enthusiastically accepted and shared the corrections proposed by the reviewers, we have answered their questions and modified the text of the manuscript, as recommended by them. In particular we have:

Reviewer 1.

A narrative review about current perspectives in head and neck tumor management. I found the article quite interesting to read,; I have some queries:

  • Although the article is not a systematic review, It would be useful to add a materials and methods section, where authors should highlight what databases were searched (PubMed, Scopus, Google Schiolar) and what keywords were used, even if the subsequent selection of the articles was not sistematical.
  1. r) we have added a section entitled “matherials and methods” as suggested by the reviewer

  • I think it should be at least reported in the manuscript's introduction that: " Various topical drugs have been proposed in the management of HNSCC and as an adjuvant or neoaudivant therapy, with controversial results" and cite an article such as : doi: 10.3390/medicina57060563. and doi: 10.3390/curroncol28040213.
  1. r) we have added the statement, as suggested by the reviewer, but we are not able to insert the two references in the bibliography list, so we have added them in the brackets (ref 2 and 3)

  • What is said in the conclusion paragraph is very interesting...i think the authors should expand it and add their personal considerations. You should also move figure 1 earlier in the text.
  1. r) we have expanded the “conclusion” paragraph as suggested by the reviewer

We have used the Track n changes system for correction

Best Regards

Dr Francesco Perri, Corresponding author

Naples, 11/8/2021

Reviewer 2 Report

Dear Editor,

I carefully read the manuscript by Fasano et al.

My comments and suggestions are the following:

  • English language needs to be carefully revised, in order to correct the typos (also in the Affiliations). Furthermore, the language is now very informal and it is not definitely adequate for a scientific article.
  • I suggest the authors not to use abbreviations as keywords.
  • Line 28: A reference is missing here.
  • Line 31: A reference is missing here. In general, authors did not include any reference in the Introduction... this is unusual and also formally wrong.
  • Figure 1 should be included in the main manuscript instead of the conclusions.
  • Authors should consider to include a summary table (also in order to make the manuscript more citable by other researchers).

Author Response

Dear Editor,

We have enthusiastically accepted and shared the corrections proposed by the reviewers, we have answered their questions and modified the text of the manuscript, as recommended by them. In particular we have:

Reviewer 2.

I carefully read the manuscript by Fasano et al.

My comments and suggestions are the following:

  • English language needs to be carefully revised, in order to correct the typos (also in the Affiliations). Furthermore, the language is now very informal and it is not definitely adequate for a scientific article.
  1. r) we delivered the manuscript to a person with high experience in English editing, as suggested by the reviewer
  • I suggest the authors not to use abbreviations as keywords.
  1. r) we have deleted the abbreviations from the keywords

3) Line 28: A reference is missing here.

  1. r) we have added the reference

  • Line 31: A reference is missing here. In general, authors did not include any reference in the Introduction... this is unusual and also formally wrong.
  1. r) we have added more references in the “introduction” chapter, as requested by the reviewers

      5) Figure 1 should be included in the main manuscript instead of the conclusions.

  1. r) we have not followed this recommendation but we have better explained it in the “conclusions” paragraph

      6) Authors should consider to include a summary table (also in order to make the manuscript more citable by other researchers).

  1. r) we have added the table 1 with this aim

We have used the Track n changes system for correction

Best Regards

Dr Francesco Perri, Corresponding author

Naples, 11/8/2021

Reviewer 3 Report

This review has the aim to summarize all the present knowledge about H&N new therapeutic options. The text is complete and practical, vithe the correct citations.

I'll suggest the authors an extense revision of english language, to make the paper easier to read. 

Author Response

Dear Editor,

We have enthusiastically accepted and shared the corrections proposed by the reviewers, we have answered their questions and modified the text of the manuscript, as recommended by them. In particular we have:

Reviewer 3.

This review has the aim to summarize all the present knowledge about H&N new therapeutic options. The text is complete and practical, vithe the correct citations.

  • I'll suggest the authors an extense revision of english language, to make the paper easier to read.
  1. r) we delivered the manuscript to a person with high experience in English editing, as suggested by the reviewer

We have used the Track n changes system for correction

Best Regards

Dr Francesco Perri, Corresponding author

Naples, 11/8/2021

Reviewer 4 Report

The paper is an interesting review describing the therapeutic options for head and neck squamous cell carcinoma (HNSCC). The description of past, present and future strategies from a translational point of view provides a good therapeutic scenario of HNSCC. The work is well written and structured in all parts. The bibliography contains useful information on the topic. 

Author Response

Dear Editor,

We have enthusiastically accepted and shared the corrections proposed by the reviewers, we have answered their questions and modified the text of the manuscript, as recommended by them. In particular we have:

Reviewer 4.

The paper is an interesting review describing the therapeutic options for head and neck squamous cell carcinoma (HNSCC). The description of past, present and future strategies from a translational point of view provides a good therapeutic scenario of HNSCC. The work is well written and structured in all parts. The bibliography contains useful information on the topic.

We have used the Track n changes system for correction

Best Regards

Dr Francesco Perri, Corresponding author

Naples, 11/8/2021

Round 2

Reviewer 1 Report

The authors responded to all queries. The paper is publishable.